# Cancer Metabolism: Fasting Reset, the Keto-Paradox and Drugs for Undoing

**DOI:** 10.3390/jcm12041589

**Published:** 2023-02-17

**Authors:** Maurice Israël, Eric Berg, Guy Tenenbaum

**Affiliations:** 1Institut Alfred Fessard, CNRS, 2 Av. Terrasse, 91190 Gif-sur-Yvette, France; 2Independent Researcher, 4501 Ford Ave., Alexandria, VA 22302, USA; 3Independent Researcher, 5558 E Leitner Drive, Coral Springs, FL 33067, USA

**Keywords:** SCOT-ACAT1 inhibitions, tumor ketolysis, acetohydroxamates, siderophores, karrikins, butenolides

## Abstract

In tumor cells, ketolysis “via” succinyl-CoA: 3-oxoacid-CoAtransferase (SCOT) and acetyl-CoA acetyltransferase 1 (ACAT1) is a major source of mitochondrial acetyl-CoA. Active ACAT1 tetramers stabilize by tyrosine phosphorylation, which facilitates the SCOT reaction and ketolysis. Tyrosine phosphorylation of pyruvate kinase PK M2 has the opposite effect, stabilizing inactive dimers, while pyruvate dehydrogenase (PDH), which is already inhibited by phosphorylation, is acetylated by ACAT1 and is doubly locked. This closes the glycolytic supply of acetyl-CoA. In addition, since tumor cells must synthesize fatty acids to create new membranes, they automatically turn off the degradation of fatty acids into acetyl-CoA (“via” the malonyl-CoA brake for the fatty acid carnityl transporter). Thus, inhibiting SCOT the specific ketolytic enzyme and ACAT1 should hold back tumor progression. However, tumor cells are still able to take up external acetate and convert it into acetyl-CoA in their cytosol “via” an acetyl-CoA synthetase, which feeds the lipogenic pathway; additionally, inhibiting this enzyme would make it difficult for tumor cells to form new lipid membrane and survive.

## 1. Introduction

Glucose and ketone bodies are the two major sources of nutrients for cells. When food is scarce, a mobilization of nutritional reserves in several organs will liberate or synthesize these nutrients. Fasting and starvation elicit the release of catabolic hormones: glucagon from the endocrine pancreas, epinephrine and cortisol from adrenals. These hormones act on different organs: glucagon for the liver, muscle adipose tissues and liver for the others. Catabolic hormones trigger glycogenolysis in liver and muscle, making glucose available; these hormones will activate the synthesis of glucose through neoglucogenesis in the liver, while muscle proteolysis provides amino acids that support the process. On the other hand, tissues responding to anabolic hormones, insulin and insulin-like growth factor (IGF) will take up glucose. Glycolysis leads to pyruvate, recovering part of the energy of glucose. Then, mitochondrial oxidative metabolism takes over. This process is energetically more efficient than glycolysis, extracting energy through the citric acid Krebs cycle, which is coupled with the respiratory electron transport chain. The citric acid cycle begins with the conversion of pyruvate into acetyl-CoA by an enzyme pyruvate dehydrogenase (PDH), controlling the glycolytic entry in oxidative metabolism. 

The production of ketone bodies in the liver requires the mobilization of lipid stores; catabolic hormones trigger lipolysis, which provides fatty acids. These acids enter mitochondria through a carnityl-driven transporter, and the beta-oxidation pathway cuts them into two carbon units, forming acetyl-CoA. Normal liver is ketogenic, converting acetyl-CoA into ketone bodies; no ketolysis takes place in normal liver, which releases ketone bodies such as beta-hydroxybutyrate (BHB) in the blood. Ketolysis will then support the metabolism of tissues responding to anabolic hormones; primarily insulin and IGF. Three enzymes form the ketolytic pathway, which converts BHB into acetyl-CoA. The only specific ketolytic enzyme is (SCOT), the product of the OXCT1 gene. The acetyl-CoA coming from ketone bodies feeds the ketolytic entry into the citric acid cycle. 

In fasting, the mobilization of tissue reserves supports the energetic and metabolic demands of active mitotic tissues through the supply of glucose or ketone bodies. 

## 2. The Metabolic Vulnerability of Tumor Cells

### 2.1. Tumor Metabolism 

We know since Warburg, Eigenbrodt and Mazurek made major discoveries surrounding cancer [1,2,3,4] that tumor cells display an inhibition of the last step of glycolysis, the pyruvate kinase M2 isoform (PK M2), providing poorly active dimers that predominate overactive tetramers. Thus, phosphorylation inhibits the enzyme. In order to overcome this “bottleneck”, tumor cells must incorporate much more glucose when making pyruvate and draw energy from glycolysis. However, there is a second “bottleneck” at the pyruvate dehydrogenase (PDH) entry in the mitochondria, which triggers the citric acid Krebs cycle. A phosphorylation also inhibits PDH, which fails to convert pyruvate into acetyl-CoA. Hence, the excess pyruvate ferments into lactic acid, even in the presence of oxygen (Warburg effect). The release of lactic acid through monocarboxylate transporters MCT1-4 decreases the intracellular concentration of lactic acid, attenuating the end-product inhibition of lactate dehydrogenase (LDH). This enzyme will then be able to convert pyruvate into lactate and NADH into NAD+. The production of NAD+ is necessary for glycolysis to proceed at the glyceraldehyde phosphate dehydrogenase (GAPDH) step. Preserving this ATP producing step, which requires NAD+, is essential for tumor survival since the other ATP producing step at PK M2 works at a low rate, forming a first “bottleneck” at the end of the glycolytic pathway, before a second PDH “bottleneck” at the entry in oxidative metabolism. We also know that the citrate condensation reaction starting the citric acid Krebs cycle increases in tumor cell mitochondria; this will feed the synthesis of fatty acids and lipids required to form new membranes for mitotic tumor cells [5]. The exit of citrate from mitochondria starts the lipogenic pathway in the cytosol via ATP citrate lyase (ACL) and acetyl-CoA carboxylase (ACC). The product of ACC is malonyl-CoA; the latter automatically turns off the mitochondrial carnityl-transporter of fatty acids and stops their degradation through beta-oxidation into acetyl-CoA when the synthesis of fatty acids is operational [6], which is indeed the case in mitotic tumor cells. Evidently, tumor cells need to acquire their mitochondrial acetyl-CoA to support their elevated citrate condensation. Since both the glycolytic and fatty acid acetyl-CoA entries in oxidative metabolism are narrow, tumor cells have to use ketones to create their mitochondrial acetyl-CoA through ketolysis. Liver ketogenesis forms and releases (BHB) in the blood; it enters tumor cells through monocarboxylate transporters MCT1-4, and then a BHB dehydrogenase forms acetoacetate, one of the SCOT substrates, while the other SCOT substrate, succinyl-CoA, comes from alpha-ketoglutarate dehydrogenase and from glutamate dehydrogenase fed by elevated glutaminolysis. The enzyme SCOT transfers the CoA of succinyl-CoA to acetoacetate, producing acetoacetyl-CoA. Then, the thiolase ACAT1 creates acetyl-CoA, which feeds the elevated citrate condensation reaction of the Krebs cycle [7]. 

### 2.2. An Essential Observation Illustrates the Ketolytic Dependency of Tumor Cells

The enzyme ACAT1, just downstream of SCOT, forms active stable tetramers in tumor cells (tyrosine 407 phosphorylation); this triggers the SCOT reaction and the synthesis of acetyl-CoA through ketolysis. Moreover, the tumor cells will stabilize this vital ketolytic entry because ACAT1 tetramers acquire an acetylation capability, capable of further acetylating and inhibiting the PDH entry, which is already inhibited by phosphorylation; this closes the glycolytic entry in the citric acid cycle and strengthens the ketolytic dependency of the tumor cell [8]. We also know that BHB is a histone deacetylase (HDAC) inhibitor, preserving the acetylation of histones, which induces the expression of fetal genes; for example, fetal hemoglobin in sickle cell anemia in which there is a mutation of the adult hemoglobin gene. We found that this was true for many other diseases, in which BHB induces, together with other factors, the expression of the fetal gene to replace an adult mutated gene [9]. Regarding cancer, the PK M2 is a fetal splice variant of the adult enzyme. 

The influx of BHB through monocarboxylate transporters (MCT1-4) [10] will form acetoacetate in the cell “via” a BHB dehydrogenase and feed the supply of mitochondrial acetyl-CoA “via” SCOT-ACAT1. On the other hand, BHB dehydrogenase converts NAD+ into NADH, counteracting the effect of lactate release that forms NAD+. This is beneficial to tumor GAPDH. An inhibition of SCOT-ACAT1 should increase acetoacetate, which would inhibit BHB dehydrogenase (end-product inhibition). As a result, less BHB would feed mitochondrial ketolysis and more BHB would support the epigenetic effects in the nucleus by inhibiting HDAC, which keeps histones acetylated; this induces the expression of fetal proteins. Other ligands of the adult protein are also involved. The epigenetic effect induces a “stem cell” phenotype in which mitotic cells fail to differentiate. In this case, the best course of action may be to inhibit MCT1-4 transporters for lactate efflux and BHB influx; the lactic acid accumulation would then block LDH, impairing the supply of NAD+ and tumor glycolysis, while the decrease in BHB influx would starve the tumor cell, impairing the supply of mitochondrial acetyl-CoA. The decrease in the BHB influx would stop the inhibition over HDAC. Histones would then be deacetylated, which favors the expression of adult proteins and differentiation, and the apoptosis of tumor cells would become possible, particularly if one inhibits SCOT and ACT1 on the ketolytic pathway, which represents the main supply of acetyl-CoA to tumor mitochondria. Slowing the citrate condensation reaction, the exit of citrate from mitochondria, starting the lipogenic fatty acid synthesis impairs the synthesis of new membranes for mitotic cells. In previous works, we tested lipoic acid and hydroxycitrate in pursuit of this goal; more information can be found in Refs. [11,12]. We found that, respectively, these substances inhibit the citrate formation and ACL, beginning the cytosolic synthesis of fatty acids. However, tumor cells are still able to incorporate exogenous acetate, converting it into acetyl-CoA in the cytosol; “via” acetyl-CoA synthetase, we can inhibit this entry with allicine [13] or orotic acid [14]. 

One observation from a marine biology perspective is particularly interesting. A compound with a butenolide furan ring was extracted from a marine bacterium Streptomyces, the Butenolide (5-octylfuran-2(5H)-one), maintains the larval phenotype of Barnacles (Amphibalanus amphitrite), delaying their fixation on ships or rocks and their transition to adult form; this antifouling effect seems related to ACAT1 and ketolysis [15]. Presumably, the ACAT1 inhibition caused by the butenolide leaves more BHB for epigenetic HDAC inhibition, prolonging the swimming larval life, and delays the settlement of swimming larva and transition to the fixed adult barnacle. Compounds such as Cochliomicin A inhibit the larval settlement through parallel activation of the NO-guanylate cyclase pathway [16]. 

A similar observation comes from vegetal biology: plants secure their access to light and nutrients by suppressing their close competitors. They achieve this by releasing HDAC inhibitors in the rhizosphere. These inhibitors (Benzoxazinoids), for example DIBOA, are cyclic hydroxamic acids structures, which are further degraded into more stable compounds with HDAC activity [17] limiting the development of competitors, presumably through some similarities with indole auxins. 

In general, HDACs, together with other compounds (NO donors), control the embryonic to adult transition for many protein isoforms. This depends, at the epigenetic level, on metabolites formed in glycolytic-ketogenic-ammonotellic fetal metabolism, or metabolites formed in adult oxidative metabolism. In mammals, a series of proteins of the “aquatic fetus” thus adapt to life in an environment with air and gravity [9]. 

Another observation relates to butenolides controlling the germination of dormant seeds after forest fires. The smoke, butenolides (karrikins, striglol) [18], signals to the seed that it can germinate: “if there was fire—there is air”. Presumably, karrinins mimic plant hormones such as auxin and gibberellerin. ACAT1 inhibition by butenolides may cancel the acetylation blocking PDH [8] and turn on the oxidative metabolism. 

It may be interesting to determine if vitamin C, which is a butenolide inhibits ACAT1, explaining some of its favorable effects in cancer treatment. 

### 2.3. What about Ketone Diets?

A ketolytic-dependent tumor will grow if fed with BHB, suggesting that ketogenic diets might be inappropriate. However, other observations show that fasting ketones would be beneficial, which may not be the case for high fat ketogenic diets. This deserves an explanation. It has been determined that saturated fatty acids of 16 carbons and above, or trans-unsaturated ones accompanied by high fat diets, stimulate AMP deaminase [19]. The resultant decrease in AMP will then reduce the stimulation of AMP kinase and block the inhibitory action of AMP kinase on ACC, which becomes active, boosting the lipogenic pathway. An increase in ketolysis occurs in order to produce the necessary acetyl-CoA and citrate. The increase in lipogenesis will then support the synthesis of lipidic membranes for mitotic cells. During fasting, this lipogenic boost by saturated fatty acids does not occur. On the contrary, if unsaturated components such as docosahexaenoic acid (DHA) are provided, they will inhibit AMP deaminase and keep AMP kinase active, inhibiting ACC and lipogenesis, impairing the formation of new membranes.

Suppose that one inhibits the ketolytic pathway with SCOT and ACAT1 inhibitors, and eventually blocks the MCT2-4 transporters. The tumor should starve; moreover, there are observations indicating that decreasing intracellular BHB interrupts the stimulation of NFkB-mediated transcription and inflammation. Extracellular BHB will then activate its membrane receptors (HCA2), which are Gi coupled receptors, decreasing the inflammation further. Once activated by BHB or other agonists, such as niacin or dimethylfumarate, this receptor inhibits the NFkB triggered transcription of inflammatory cytokines; there is apparently an inhibition of IkB kinase, which then retains NFkB bound to IKB in the cytosol, unable to reach the nucleus. Agonists of HCA2 such as niacin are activators of prostaglandin PGD2 release from cells infiltrating the tumor. Niacin (Vitamin B3) and PGD2 elicit a skin flush [20].

In summation, intracellular BHB transported by MCT 1-4, will support tumor nutrition by ketolysis and NFkB-associated inflammation, while extracellular BHB signaling through HCA2 receptors retains the transcription factor NFkB in the cytosol, inhibiting inflammation. This may explain opposite observations on the effects of BHB on tumors [21,22].

In the same vein, other works using breast cancer xenografts show that ketones and lactate “fuel” tumor growth and metastasis [23], while other groups show that ketones decrease tumor cell viability and prolong survival using VM3 mice with metastatic cancer [24]. The MCT transporters can operate in both directions releasing lactic acid if its cellular concentration is elevated or incorporate BHB to fuel the ketolytic pathway. By inhibiting lactate dehydrogenase, one changes the lactic acid concentration gradient. Additionally, an increase in lactic acid in tumor cells induces oxidative stress and, presumably, intracellular acidification, which inhibits tumor progression [25]. The reciprocal interaction of tumor cells with cancer-associated fibroblasts is of particular interest; apparently, during contact with prostate cancer cells, fibroblasts reprogram their metabolism. They take up more glucose and release more lactate by expressing new MCT4 transporters. Reciprocally, tumor cells shift toward an aerobic metabolism; they decrease their glucose transporters and increase the uptake of lactate via the lactate transporter MCT1. They adapt to use lactate for their anabolism. A symbiotic tumor–stroma interaction develops and seems to depend on the hypoxia-inducible factor, HIF1. The pharmacological inhibition of MCT1-mediated lactate upload affected these reprogrammed cancer cells able to use lactate for their anabolism and survival [25,26].

## 3. Metabolic Features of Ketolytic-Dependent Tumors

### 3.1. The Keto Diet: Is It Appropriate?

An early observation by Brünings, recalled by Klement [27], deserves to be mentioned. In his study, patients were given a low-carbohydrate diet, which was associated with an insulin injection protocol that aimed to decrease the supply of glucose to tumors. Initially he observed a decrease in tumor sizes, and later a rebound, while feeding patients with a high-fat diet. We now know that insulin elicits the release of somatostatin from delta cells, which decreases glucagon and ketogenesis leading to a probable decrease in BHB and growth of ketone-dependent tumors. However, since Brünings fed patients with a high-fat ketogenic diet, a rebound of tumors took place. Brünings was not aware that somatostatin release triggered by the insulin injections was probably behind the initial reduction in tumors [28]. Indeed, somatostatin decreases glucagon and ketogenesis at the beginning of the trial; however, a late rebound of tumors probably occurs when insulin fails to maintain somatostatin release. In time, the decline in somatostatin stops inhibiting glucagon, which then boosts ketogenesis, providing ketones to the tumor [29]. In an earlier work on animal cancer models, we observed that octreotide, a somatostatin analog, decreases tumor volumes [12].

In previous publications, we tried to explain the metabolic rewiring of tumor cells by considering that tumor cells respond to both anabolic insulin and to catabolic glucagon, since enzymes such as PK and PDH remain phosphorylated, as for glucagon-mediated neoglucogenesis. Meanwhile, glycolysis and anabolism elicited by insulin affect tumor cells displaying a hybrid metabolic situation. We have incriminated a deficient regulation of the endocrine pancreas, in which the release of GABA by beta cells failed to turn off alpha cells releasing glucagon and delta cells releasing somatostatin. The failure of this mechanism sends a dual anabolic/catabolic signal perceived by new stem with both receptors active, while differentiated cells desensitize their insulin receptors because the GABA failure fails to turn off insulin release. Concurrently, constant leakage desensitizes insulin receptors for differentiated tissues, which is not the case for new mitotic cells with new insulin receptors. Hence, these cells rewire their metabolism in the hybrid mode, while differentiated tissues produce glucose and ketone bodies to feed the ketolytic-dependent tumor mitotic cell.

### 3.2. Why Is Pyruvate Kinase Phosphorylated?

In neoglucogenesis elicited by glucagon, the hormone acts on Gs coupled receptors, stimulating adenylate cyclase and forming cAMP. The latter controls the direction of the glycolytic pathway by inhibiting an activator of glycolysis, which also stimulates neoglucogenesis: Fructose 2-6 bis Phosphate. This will stop glycolysis and stimulate neoglucogenesis. The glucagon receptor activates protein kinases A (PKA), then Src tyrosine kinase that inhibits pyruvate kinase (PK). This is necessary because PK works only in the PEP to pyruvate direction. Inhibiting PK via phosphorylation avoids returning to pyruvate, allowing PEP to feed and support the synthesis of glucose. Pyruvate forms through transamination of amino acids, then pyruvate carboxylase gives OAA in mitochondria and PEP carboxykinase feeds PEP in the cytosolic neoglucogenic pathway.

### 3.3. How Do We Switch Back to Glycolysis?

Muscle activity, or insulin action through IP3 (inositol 1,3,5 phosphate) increases intracellular calcium. This activates a phosphodiesterase converting cAMP into AMP, which cancels the inhibition of cAMP over Fructose 2-6 bis P; this activates glycolysis and deactivates neoglucogenesis. Calcium will also stimulate calcineurin phosphatase, which dephosphorylates an inhibitor I1 of PP1 phosphatase; it will then dephosphorylate and activate PK and PDH. Moreover, calcium will increase the membrane incorporation of glucose transporters, reactivating glycolysis.

#### 3.3.1. Selection of the Acetyl-CoA Source

The increase in AMP activates AMP kinase, which inhibits ACC and fatty acid synthesis, the decrease in malonyl-CoA, unlocking the carnityl-transporter of fatty acid and their degradation into acetyl-CoA. Since lipogenesis is blocked, DAG does not form triacylglycerol (TAG). The increase in DAG will then stimulate Protein kinase C (PKC), triggering the formation of an inhibitor (CPI 17) of PP1 phosphatase. This maintains the phosphorylation of PK and PDH by kinases, which closes the glycolytic acetyl-CoA supply, while the fatty acid acetyl-CoA source is operational. On the other hand, if DAG decreases, PKC stops forming the CPI 17 inhibitor of PP1 phosphatase; consequently, both the glycolytic and fatty acid supplies of acetyl-CoA are operational.

AMP + DAG+: fatty acids give acetyl-CoA; the glycolytic acetyl-CoA supply is closed.

AMP + DAG−: both fatty acid and glycolysis can provide acetyl-CoA.

#### 3.3.2. Role of AMP Deaminase

We have seen that saturated fatty acids of 16 carbons, accompanied by high-fat diets, stimulate AMP deaminase. The resultant decrease in AMP inhibits AMP kinase, canceling its inhibitory action over ACC, which is activated. This boosts the fatty acid synthesis pathway, while malonyl-CoA turns off the fatty acid degradation into acetyl-CoA. If DAG decreases (and converts to TAG) it will not stimulate PKC. Thus, the CPI 17 inhibitor of PP1 does not form, and the phosphatase dephosphorylates and activates PK and PDH, opening the glycolytic supply of acetyl-CoA. On the contrary, if DAG increases (following the action of Growth hormone, for example) PKC is stimulated, forming the CPI 17 inhibitor of PP1, which maintains the phosphorylation of PK and PDH. Since both the fatty acid and glycolytic acetyl-CoA supplies are closed, tumor cells must then use ketone bodies from the liver to create their acetyl-CoA.

AMP − DAG−: fatty acid supply of acetyl-CoA closed; glucose provides acetyl-CoA.

AMP − DAG+: both glycolytic and fatty acid acetyl-CoA sources are closed; ketone bodies provide acetyl-CoA (as in tumor cells).

### 3.4. A Metabolic Hypothesis for How Cancer Forms?

Many internal or external causes can elicit cell death, triggering the mitosis of stem cells for tissue repair. If there is a chronic lesion of the endocrine pancreas affecting the anabolic–catabolic mutual exclusion mechanism, which depends on the transmitter GABA, then a dual message reaches stem cells that rewire their metabolism and become dependent of ketone bodies, while insulin-desensitized differentiated cells respond to the catabolic part of the message, providing the ketones to mitotic cells. Their ketolytic metabolism has epigenetic effects inhibiting differentiation of mitotic daughter cells, leading to a geometric increase in the tumor.

## 4. How to Undo the Metabolic Advantage of Tumors

The peculiarities of tumor cell metabolism that might be taken advantage of for cancer treatment have been discussed in works that consider the alterations of signal transduction pathways and metabolic reprogramming of tumor cells [30]. A common feature of tumors is to acquire nutrients, in spite of their specific metabolic alterations, and build their biomass. Associated metabolites induce epigenetic effects on tumor cells and their microenvironment [31]. In addition, the Warburg effect was revisited in relation to oxidative metabolism, indicating that mitochondria and oxidative metabolism play a prominent role in cancer, with flexible adaptation, when one controls the Warburg effect [32]. 

### 4.1. A Possible Reset of the Metabolic Rewiring Mechanism in Cancer

We have seen that tumor cells avidly consume glucose, but two bottlenecks at PK M2 and PDH cut the entry in oxidative metabolism. This is similar to the process that occurs if a phosphorylation inhibiting these enzymes does not reverse while switching from neoglucogenesis to glycolysis. If one triggers neoglucogenesis through a period of fasting, in which these phosphorylations normally occur, one may try to reset the system at the interruption of the fasting period in the hope that the phosphatases will correct the anomaly, which maintains them in the OFF configuration. This switching mechanism, involving the role of calcium, a PDE and a phosphatase such as calcineurin, was discussed earlier in the paper. During the fasting period, one should accept complements that preserve calcium stores and the enzymes capable of resetting the system. On the other hand, fasting forms ketone bodies that feed the tumor; we have indicated that these fasting ketones are not as bad as are those from high fat ketogenic diets. The incorporation of complements such as green tea and epigallocatechin [33,34] during the fasting period would inhibit MCT2-4 transporters and the ketone influx while favoring the ketone signaling action transmitted by HCA2 receptors, which is helpful. Moreover, DHA and EPA would limit the ketone contribution to the lipogenic pathway. At the end of a long fasting period, one observes some favorable cases; they might result from a successful reset, removing the abnormal phosphorylation blocking the enzymes PK, M2 and PDH.

### 4.2. Cutting the Ketone Influx

In Figure 1, we recall the inhibition of both the glycolytic and fatty acid sources of mitochondrial acetyl-CoA indicated by PK and PDH interruptions on the glycolytic entry, while a single interruption by malonyl-CoA cuts the fatty acid source of acetyl-CoA, since the tumor must synthesize its fatty acids to create new membranes. Hence, the ketolysis of BHB becomes a major acetyl-CoA supply, promoting, for example, the growth of breast tumors. Adipocytes form BHB, which enters tumor cells through MCT 2 transporters [35,36].

One should also control the BHB influx. Indeed, once in the cell, BHB has multiple effects; first, it feeds the SCOT pathway; second, it goes to the nucleus, where it inhibits HDAC, favoring the acetylation of histones, which induces the expression of fetal genes. A third effect of intracellular BHB is the activation of NFkB-mediated transcription of inflammatory cytokines Il1B. Indeed, intracellular BHB induces phosphorylation of IKB, liberating the NFkB transcription factor, which moves to the nucleus. Thus, if one inhibits the influx of BHB with compounds such as Syringopine or Syrosingopine from Rauwolfia [36] or others such as Quercetin or Epigallocatechin [33,34] tumor cells will starve, since ketolysis should decline, particularly if associated with SCOT and ACAT1 inhibitors. In parallel, the decrease in intracellular BHB stops the inhibition over HDAC, which will deacetylate histones, silencing fetal genes, and elicit a transition to adult genes. Hydroxamic acid HDAC derivatives, which display anticancer effects, are probable SCOT inhibitors, and may be able to starve the tumor. Finally, the decrease in intracellular BHB stops the stimulation of inflammatory cytokines transcription mediated by NFkB, since NFkB remains bound to IkB in the cytosol.

In summation, the MCT transporter of BHB driven by SCOT-ACAT1 attracts this essential nutrient for tumor cells. In parallel, this elicits NFkB transcription and inflammation, whereas HDAC inhibition by BHB elicits histone acetylation and the expression of fetal genes. By inhibiting the MCT transporter of BHB and the SCOT-ACAT1 pathway, one starves the tumor and blocks NFkB transcription; histones are deacetylated by HDAC, favoring a shift to adult genes. Moreover, extracellular BHB activates HCA2 receptor signaling as other agonists of these receptors, such as niacin. These receptors inhibit inflammation by keeping NFkB bound to IkB in the cytosol, further blocking inflammation and inflammatory macrophages; these effects would be mediated by prostaglandins. Niacin or other agonists of this receptor should decrease inflammation and NFkB-mediated transcription, as adrographolide (Andrographis Paniculata) [37].

In conclusion, the ketone body is favorable if it stays outside the cell, acting as a signaling molecule rather than as a nutrient for the tumor. This summarizes the keto paradox and explains some opposing observations on the effects of BHB on tumor cells.

### 4.3. Cutting the SCOT-ACAT1 Ketolytic Steps

Our strategy is to block the ketolytic pathway, particularly the specific step driven by SCOT and the supply of acetyl-CoA to lipogenesis, as we recently attempted on an animal cancer model [38]. Moreover, since active ACAT1 tetramers collaborate with SCOT to pull in the ketolytic nutrient, we suggest using inhibitors of both enzymes.

A typical inhibitor for SCOT is acetohydroxamic acid, which was introduced by Pickart and Jencks [39]. There are many hydroxamic acid derivatives in the Siderophores iron chelator family molecules, and it would be interesting to test them on SCOT activity. An association with ACAT1 inhibitors, Butenolides, including vitamin C, Arecoline and trimetazidine (Vastarel) discussed above would certainly be interesting to test.

### 4.4. Possible Compounds for Inhibiting SCOT

Lithostat acetohydroxamic acid is a typical SCOT inhibitor used to treat bladder stones. Another inhibitor, Pimozide [40,41], used to treat mental diseases, reduces cancer incidence. Several interesting compounds are highlighted in the work of Lissanti’s group, who describe potential SCOT inhibitors, the Mitoketoscine [42], through their structural and binding properties. Selecting the best and least toxic derivative for animal cancer models requires collaboration with pharmaceutical groups.

We can also explore the literature for hydroxamic acid derivatives given for other pathologies; this is the case for Vorinostat SAHA, or Quisinostat and Trichostatin, which are HDAC inhibitors [43] that display anticancer properties; it would be interesting to see if they inhibit SCOT as Lithostat. Another way to find potential SCOT inhibitors comes from bacteriology. In order to survive in a hostile environment such as in stomach acid or in the bladder, the bacteria Helicobacter or Proteus use their urease to make ammonia and neutralize the acidity. Drugs such as acetohydoxamic acid inhibit the urease by binding the metal (Ni) in the active center, blocking the alkalinization of urine and stone formation. Another source of compounds used to find SCOT inhibitors is the pharmacology of Metalloproteinase inhibitors; again, the hydroxamate derivatives that block the Zn-metalloproteinases deserve to be studied in relation to SCOT inhibition [44]. The iron-binding siderophores of bacteria, algae and other organisms are likely the best to explore. In our aerobic environment, iron bioavailability is low due to its insolubility. Iron is necessary for the survival of these organisms; thus, they synthesize iron chelators to capture ferric iron. One category of these siderophores is hydroxamate derivatives; they are potential SCOT inhibitors and have clear anticancer properties [45]. We recently compared acetohydroxamic Lithostat and Lithothamnion from red algae Lithothamnion calcareum [46] on the Lewis cancer model and confirmed observations on their anticancer properties [38]. An interesting siderophore is Ferrichrome from Lactobacillus casei, which inhibits colon cancer [47]. The iron chelators desferroxamine (desferal) display antineuroblastoma effects [48], and as such they would be interesting for future leukemia treatments. Soluble seawater marine siderophores (Ferrioxamines) phytoplankton siderophore from Rhodomonas Ovalis or siderophore from the marine bacterium Alteromonas luteoviolacea (alterobactine) [49] would be interesting to test using the SCOT assay developed by Williamson [50,51]. The problem is finding the best SCOT inhibitor that would reach the enzyme without being toxic. Presumably, a hydroxamate derivative originating from the biology of siderophores or a compound related to iron metabolism would do the job. We know that the normal liver never expresses the specific ketolytic enzyme SCOT; only ketogenesis takes place in liver. Is this related to the recycling of the heme, iron recovery and biliverdin, bilirubin metabolism in the liver?

During bacteria or parasite infections, the microorganisms try to rob our iron with their siderophores, and proteins of our immune system with better affinity for iron compete with bacterial siderophores.

A colleague pointed out that melatonin has reactive groups similar to acetohyroxamic acid. We also notice that febrifugine from Dichoa febrifuga used in Chinese medicine and its derivative Halofuginone [52] have similar reactive radicals; they both have anticancer effects, affinity for iron and might inhibit SCOT.

### 4.5. Compounds Affecting ACAT1

ACAT1 activity in association with SCOT secures the vital ketolytic supply to tumor cells by pulling in the influx of BHB. The antifouling Butenolide, or Vitamin C, or Karrikins and strigol from smoke, would inhibit ACAT1. Arecoline from Areca catechu is an ACT1 inhibitor, but it might have oral carcinogenic effects. Other ACAT1 inhibitors (Vastarel) could be tested in this context in association with SCOT inhibitors. The acetogenin from anones are particularly interesting, since they have a furan ring. As butenolide, they might inhibit ACAT1 and inhibit AMP deaminase, which stimulates AMP kinase that cuts the lipogenic supply. Fungal siderophores from Phellinus linteus or Phellinus baummii [53,54] are identified, and might influence SCOTand ACAT1, inoscavin has a furan ring, which likely affects ACAT1.

### 4.6. Compounds Acting down Stream of SCOT-ACAT1

In earlier works we showed that Lipoic acid and Hydroxycitrate from Garcinia were able to limit the citrate efflux from mitochondria and inhibit ATP citrate lyase (ACL) in the cytosol, which cuts the lipogenic supply. These effects are strengthened if one uses allicine or orotic acid to block the direct incorporation of external acetate via acetyl-CoA synthetase. Moreover, one can impair the synthesis of lipid membranes with DHA through the inhibition of AMP deaminase, which leaves more AMP to stimulate AMP kinase. This then inhibits ACC, at the start of the lipogenic pathway. One can reinforce these effects by decreasing the synthesis of cholesterides, using statins that are difficult to handle, or Bergamotin [55]. An interesting report [56] shows that tumors can use the glutamine entry in the citric acid cycle to form citrate and feed the lipid synthesis pathway. The role of isocitrate dehydrogenase (IDH2) is essential in this process; IDH2 mediates a reductive carboxylation to facilitate the utilization of glutamine through alpha ketoglutarate to citrate reversible steps of the Krebs cycle and feed the citrate supply to lipid synthesis. In this report, the long-term induction of the oncogene K-ras upregulates IDH2; this facilitates glutamine utilization for lipid synthesis during malignant transformation. 

There is certainly a great variability among tumors and their metabolic adaptations. The presentation given in this review did not consider the different tumor situations or the magnitude of the glycolytic bottlenecks, or the degree of the ketone dependency. 

## 5. Conclusions

With a PDH locked by phosphorylation and by acetylation, tumor cells have difficulty to use the glycolytic source of acetyl-CoA; in a way they are abnormal, “as in Beriberi”. Moreover, the PK M2 bottleneck pushes them to increase the influx of glucose fermented into lactic acid. Tumor cells must synthetize fatty acids needed for their mitotic lipid membranes; this process automatically inhibits the degradation of fatty acid into acetyl-CoA. Hence, they become dependent on ketone bodies to make their mitochondrial acetyl-CoA. The dark side of ketolysis is the nutrition and growth of tumors, an effect opposite to a beneficial ketone signaling action, which leads to contradictory results on the effect of ketones in cancer research; this is discussed in the keto paradox.

We think that the specific ketolytic enzyme SCOT is a major target for cancer treatment. However, we do not think that a single compound would be sufficient for developing a metabolic treatment. The strategy is to attack each step of the ketolytic process; this would gradually starve the tumor in preference to other tissues because, unlike other tissues, tumors are particularly dependent on the ketogenic supply for their metabolism. Downstream of SCOT-ACAT1 system, attracting the ketone, Lipoic acid, Hydoxycitrate, Allicine, Docosahexanoic acid and Bergamotin should decrease the lipogenic supply. The effect of Gallepine from Gallega officinalis on AMPdeaminase is to be considered in relation to metformin. Upstream of SCOT-ACAT1, one can inhibit the ketone transporters MCT2-4 using several compounds—epigallocatechin, syringopine or syrosingopine—while ketogenesis would be slowed with octreotide, a somatostatin analog. As for SCOT, we have still to find the best hydroxamic acid derivative in the list of siderophores. We do not think that a toxic or irreversible compound is an option, since we want to selectively starve the tumor and not affect other tissues. Suggestions could be a hydroxamic acid derivative, Lithostat, Lithothamnion or even Melatonin, Ferrichrone or Febrifugine. There are many to test using the SCOT assay, with compounds coming from the field of metalloproteinases or HDACS. For ACT1, which facilitates SCOT activity, we discussed several possibilities; vitamin C, Butenolide and acetogenins, those with a furan ring, might inhibit ACAT1 [57] and stimulate AMP kinase. Other ACAT 1 inhibitors, such as Vastarel, Karrikin, Strigol and Arecoline, should be checked for side effects.

Finally, if we limit the ketogenic supply to the SCOT ketolytic pathway it could be useful to preserve the signaling action over the HCA2 receptors, activated by agonists such as niacin.

In the course of a “fasting reset” aiming to correct enzymatic anomalies, one would take these compounds after checking the mixtures on animal models for possible toxic interactions. One would certainly avoid a “high fat keto diet”, as explained in the keto paradox.

It is important to note, as we reach the conclusion of this paper, that a mixture of low toxicity compounds, affecting each step of the ketolytic supply to ketone-dependent tumor cells, would gradually suppress their metabolic advantage and immortality, rendering them visible to our immune system and lead them to an apoptotic end.

## Figures and Tables

**Figure 1 jcm-12-01589-f001:**
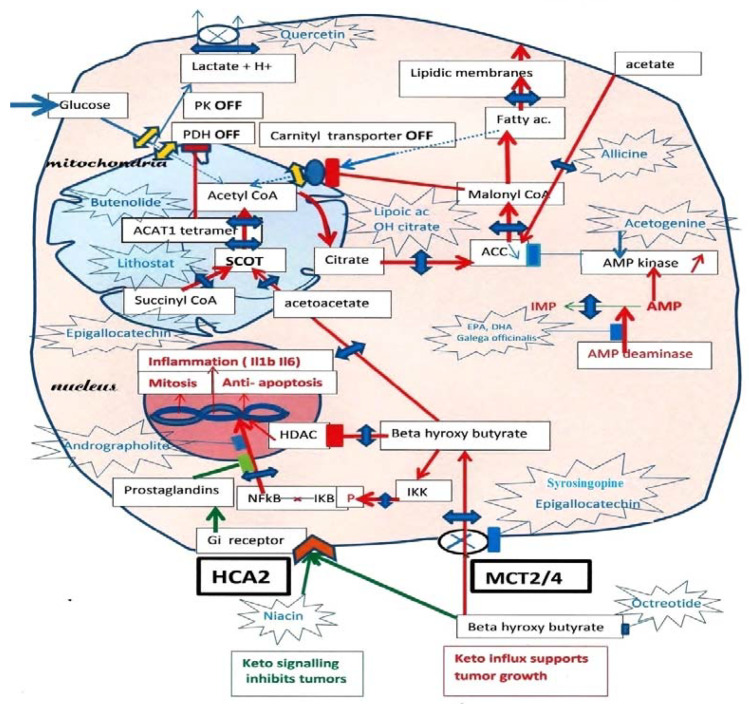
Ketolytic dependent tumor cells. Inhibitions by phosphorylation, of Pyruvate kinase PK M2, and Pyruvate dehydrogenase PDH, cut the glycolytic supply of acetyl-CoA to mitochondria, lactic acid release occurs through MCT monocarboxylate transporters. Moreover, an acetylation of PDH by ACAT1 thiolase further increases the inhibition of PDH. Since tumors must synthesize fatty acid to create new membranes, this automatically turns off the fatty acid mitochondrial carnityl transporter and the fatty acid supply of acetyl-CoA. The carnityl transporter is inhibited by malonyl-CoA, the product of Acetyl CoA carboxylase (ACC), starting the lipogenic pathway in the cytosol. Hence, ketone bodies and beta-hydroxybutyrate, (BHB) become an essential source of acetyl-CoA in tumors, via SCOT the specific ketolytic enzyme and ACAT1. Then, citrate exits the mitochondria and feeds lipogenesis via ATP citrate lyase (ACL) and acetyl-CoA carboxylase (ACC). By blocking the MCT transporters with indicated compounds, one would affect the lactate release, the influx of beta-hydroxybutyrate (BHB) and the supply of acetoacetate to SCOT. We indicate possible inhibitors for SCOT: Lithostat or other Siderophores, Ferrichrome, Phycocyanin, Febrifugine or even Melatonin. As for ACT1, Butenolides, vitamin C and others are indicated; this should starve the tumor. We suggest inhibiting the citrate condensation and supply to the lipogenic pathway using lipoic acid and hydroxycitrate, as indicated. Moreover, Allicine would cut the direct conversion of acetate into acetyl-CoA in the cytosol. An essential key is that AMP kinase will inhibit ACC, which merits stimulation by acetogenin, and to inhibit AMP deaminase with Docosahexaenoic acid (DHA) or Galega officinalis, or metformin. By inhibiting the entry of BHB, one also stops the stimulation of NFkB-mediated transcription, a hallmark of cancer, and cancels the inhibition of histone deacetylase (HDAC) caused by BHB. Upstream of ketolysis, one may use octreotide (a somatostatin analog) to decrease the ketogenic supply. There is, however, a paradox, since the signaling action of extracellular BHB on HCA2, a Gi coupled receptor, is to preserve (it inhibits NFkB transcription). A prostaglandin messenger is involved; it can be strengthened with Andrographolite. Niacin is another agonist for HCA2, which may help when using octreotide for decreasing ketogenesis. Green arrows: HCA2 signaling. Red arrows: BHB influx through MCT and effects.

## Data Availability

Not applicable.

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
