# Peer review of "Cancer Metabolism: Fasting Reset, the Keto-Paradox and Drugs for Undoing"

_jcm, 2023, doi:10.3390/jcm12041589_

Round 1

Reviewer 1 Report

I would like to thanks the authors for submitting this paper. Israel et al. tried to talk about the role of ketone bodies in oncometabolism.  Here are my major concerns:

- First, English is very poor, a lot of typos are found, it is not structured at all, the aim of this work is not displayed and not presented, we do not know what/why we are reading. What are the objectives and questions raised in this work?

- References are truely lacking. When talking about an already described phenomenon, authors is strongly advised to reference their sentences as soon as he can. Here, is it absolutely not the case, rather authors talk and described without reference - it's kind of a copy/paste review.

- Sentences are too long and do not make any sens, very difficult for us to read and follow - lacks of logic. It's strongly recommended at this stage to totally re-write this manuscript. 

- I stop here the revision as the list of negative points is becoming too long.

- Finally, I would like to state that even prominent researchers need to clearly explain their thoughts when written a review for a large audience. Otherwise, no interest to read it. Thus, I would just recommend to rewrite this manuscript and reformulate ideas that, I'm convinced, are strong and relevant.

Author Response

point 1: English editing .  I  took this service proposed by the journal , after dealing with the different points raised by the reviewer.

point 2:  I will add  the references indicated.  However, nothing  was  pasted from my previous works. The so called "keto- paradox" was not explained before. I agree that it is difficult to follow. You must consider the dual action of BHB: first as a nutrient  for ketone -dependent tumor cells, (its influx depends of MCT transporters) second, BHB is a  signaling molecule, acting on HCA2  membrane receptors,  controlling inflammation and proliferation.  Moreover, the issue is complicated by a direct epigenetic effect of BHB on HDAC inhibition.  Finally, The ketone BHB can form in fasting or after a high fat  ketogenic diets, the effects on  AMP deaminase and AMP kinase will not be the same, particularly for fatty acid oxidation and energetic metabolism.  We did an effort  to integrate all these data, sorry if you feel that I was not clear enough. Thank you for the additional references that I will comment in a specific paragraph. 

point 3:   Thank you for considering that  a" prominent researcher "should explain things better. However, it is difficult to explain a complicated issue:  particularly, if you consider that the high fat  ketone diet  is inappropriate, while fasting ketones are not too bad or  may even help.  

Please consider that inhibiting the specific ketolytic  enzyme SCOT  with  the  compounds  we suggest, may have  positive effects, as those we earlier suggested:  lipoic acid , hydoxy citrate, for backing -up  cancer treatments. 

Reviewer 2 Report

The manuscript entitled “ Cancer Metabolism: Fasting Reset, the Keto-Paradox and Drugs for Undoing” by Israël et al summarizes the cancer metabolism by particularly focusing on ketone bodies. The main body of the manuscript is interesting and highlights an important area of cancer research. However, I fell that it is not comprehensive and the major breakthrough evidences in the field somehow not discussed.  For instance, important findings in the field regarding ketone bodies and cancer such as  https://doi.org/10.1002/ijc.28809, https://doi.org/10.4161/cc.9.17.12731 , doi:10.1038/sj.bjc.6601269 ,  https://doi.org/10.4161/cc.9.17.12721 , Cancer Res (1988) 48 24_Part_1): 7264–7272 were not even discussed. Similarly, at least some of the important evidences about cancer metabolism were somehow ignored (DOI: 10.1038/nm.2492 ,  DOI: 10.1158/0008-5472.CAN-12-1949, DOI: 10.1073/pnas.1212780110 , DOI: 10.1172/JCI75836, DOI: 10.1371/journal.pone.0075154  , DOI: 10.1073/pnas.0914433107).  Therefore I feel that the manuscript is not sufficiently supported by scientific data and  falls short for to be published in the journal.  

Author Response

First: I will take the English Editing option proposed by the journal , 

Second: Answers to the scientific points.

  I know the contradiction,  illustrated by the references that the reviewer kindly  indicates . Ketones and lactate, fuel tumor growth  in some works, ,  while  opposite results  are found by others  showing  positive effects of BHB.  I will add 4 references  to the 2 others initially given, illustrating this contradiction  and comment these works, in a paragraph that  can only be written in the middle of the text.    

 The explanation proposed:  As a nutrient, the ketone BHB,  feeds the ketone- dependent  tumor, through MCT transporters , similar to those that release lactic acid. However, as a signaling molecule, acting on external HCA2 receptors, BHB has opposite favorable effects, on inflammation and proliferation.  In some works, the signaling  action mediated by  HCA2 receptors, predominates , In other works , the nutritional effect predominates  and tumors grow. 

On the other hand, BHB has epigenetic effects (it is a HDAC inhibitor). In an interesting work, an  interaction of tumor cells with local fibroblast, seems to rewire metabolic pathways in both cells, rendering the tumor cell lactate-dependent , which is  provided by local fibroblasts. This "symbiotic" epigenetic adaptation seems to be triggered by HIF1. Thank you for this reference.   

Moreover, we did explain why fasting ketones and high fat diet ketones have different effects . This depends on the action of fatty acids, coming with the high fat diet, on AMP deaminase, which controls AMP kinase and the inhibition of the mitochondrial fatty acid  transporter. This is summarized  in the  keto- paradox discussion. 

For the references on lactate dehydrogenase inhibition, and its effect on tumor cells . We considered that  MCT 1-4 transporters work in both direction , they release lactic acid,  and  ensure the influx of BHB .  Suppose  that you block MCT transporters, or inhibit LDH,  lactic ac. release  against its  concentration gradient, will decrease, an acidification  of the cell, might elicit apoptosis; or induce an oxidative stress. Whereas a decrease of the butyrate influx  impair the synthesis of acetyl-CoA in a ketolytic dependent tumor. 

As for M2 Pyruvate kinase.  I  briefly discussed it here, we earlier did this in other works, in honor of Eigenbrodt and Mazurek discoveries.  I  simply recall that M2  PK  is an embryonic splice variant of adult M1 PK  ; the expression of the M2  isoform might be related  the effect of BHB  on HDAC inhibition  in ketolytic dependent tumors.  I gave in references an early observation, showing that the effect of BHB was not limited to the expression of fetal Hemoglobin, but concerned other  fetal proteins, I gave this example to recall that M2 PK is a  fetal splice variant of PK M1.  It would be far too long to review all the works on PK M2. They are indeed essential.

I appreciate your efforts for improving the manuscript. We did have some difficulty  for  explaining opposite observations on the multiple effects of BHB.

We wish to say that it is necessary to block SCOT and ketolysis, in keto-dependent tumors and call the attention of clinicians on  potential dangers of high fat keto - diets, as in the initial observation of Brünings.

I hope that these answers incorporated in an English Edited and revised manuscript  will satisfy the reviewer . I selected 4 references in the list you kindly proposed and wrote a paragraph on them. Particularly insisting on the interaction of tumor cells with fibroblasts or other interacting cells  within the tumor. I could not select all the references. The review is is not a systematic review and is  already far too long. These references are indeed very interesting.

Round 2

Reviewer 1 Report

Thanks for having made some modifications to your text manuscript - it's now much more clear for readers. My last point would be to include these following references: 

Pavlova NN, Zhu J, Thompson CB. The hallmarks of cancer metabolism: Still emerging. Cell Metab. 2022 Mar 1;34(3):355-377. doi: 10.1016/j.cmet.2022.01.007. Epub 2022 Feb 4. PMID: 35123658; PMCID: PMC8891094.

Cassim S, Vučetić M, Ždralević M, Pouyssegur J. Warburg and Beyond: The Power of Mitochondrial Metabolism to Collaborate or Replace Fermentative Glycolysis in Cancer. Cancers (Basel). 2020 Apr 30;12(5):1119. doi: 10.3390/cancers12051119. PMID: 32365833; PMCID: PMC7281550.

Kroemer G, Pouyssegur J. Tumor cell metabolism: cancer's Achilles' heel. Cancer Cell. 2008 Jun;13(6):472-82. doi: 10.1016/j.ccr.2008.05.005. PMID: 18538731.

Author Response

Thank you for your positive comment. I have incorporated the three valuable ref. in the text. (30,31,32).  We can indeed take advantage of peculiarities of tumor metabolism for developing a metabolic treatment.  Specific Metabolic alterations induce a re-wiring of metabolic pathways, enabling tumors to acquire nutrients and build their biomass. This reprogramming is also their vulnerability. In this context even the Warburg effect was revisited in relation to oxidative metabolism. We also pointed the ketolytic dependency of tumor cell.